# Epidemiological, Diagnostic, and Clinical Features of Intracranial Cystic Echinococcosis: A Systematic Review

**DOI:** 10.3390/pathogens14121264

**Published:** 2025-12-10

**Authors:** Songul Meltem Can, Feza Irem Aldi, Muhammed Burak Sarikaya, Pelin Sari Serin, Nermin Sakru

**Affiliations:** 1Department of Neurosurgery, Acıbadem Health Group Fulya Hospital, Istanbul 34000, Türkiye; songulcan@acibadem.com; 2Department of Medical Microbiology, Faculty of Medicine, Trakya University, Edirne 22030, Türkiye; firemaldi@trakya.edu.tr (F.I.A.); pelinsari@trakya.edu.tr (P.S.S.); 3Department of Neurosurgery, Edirne State Hospital, Edirne 22030, Türkiye; mburaksarikaya@gmail.com

**Keywords:** cerebral, cystic echinococcosis, *Echinococcus granulosus*, intracranial, neurology, neurosurgery

## Abstract

Cystic Echinococcosis (CE) is a rare but serious parasitic disease caused by *Echinococcus granulosus sensu lato*, representing only 1–2% of all hydatid disease cases. Due to its nonspecific clinical presentation, its diagnosis and management pose significant challenges. This study aimed to provide a comprehensive overview of intracranial CE cases reported globally over the past 35 years, focusing on demographic characteristics, clinical presentation, diagnostic approaches, treatment modalities, and outcomes. Methods: A systematic review was conducted in accordance with PRISMA guidelines and was registered in PROSPERO (CRD 42024608624). Relevant studies published between 1990 and 2024 were identified from PubMed, Scopus, and Web of Science databases. Results: After screening and eligibility assessment, 392 studies involving 718 intracranial CE cases were included. The majority of patients were children (65%) and male (59.2%). The most frequent presenting symptoms were signs of increased intracranial pressure (79.4%), followed by motor deficits (37.9%) and visual disturbances (23.2%). Most cysts were located in the supratentorial region (88.9%), predominantly in the parietal lobe, and were solitary (88.4%). Surgical intervention was performed in 95.8% of cases, often combined with albendazole therapy. Complete recovery was observed in 85.5% of patients, while 8.7% died—primarily due to cyst rupture-related complications such as septicemia and anaphylaxis. Recurrence was reported in 26% of cases with follow-up. Conclusions: This review presents one of the most extensive analyses of intracranial CE to date. Despite being a rare manifestation, intracranial CE should be considered in the differential diagnosis of space-occupying brain lesions in endemic areas, particularly in paediatric patients.

## 1. Introduction

*Echinococcus granulosus sensu lato* (*E. granulosus s.l.*) is a parasite that causes cystic echinococcosis (CE). The definitive host is the dog, while the intermediate hosts are herbivorous mammals such as sheep and cattle, as well as humans. The parasite is found on every continent except Antarctica. In endemic regions such as Central Asia, the Mediterranean countries, China, Argentina, and Peru, incidence rates can be 50 per 100,000 people or higher. According to WHO data, more than 1 million people may be living with this disease worldwide at any given time [1,2].

The parasite mainly settles in the liver and lungs; however, the disease can also affect the kidneys, spleen, bones, and brain [1,2]. Although the involvement of these organs is less common, extrahepatic and extrapulmonary localizations often pose greater diagnostic and therapeutic challenges [3]. Intracranial CE disease accounts for just 1–2% of all cases of the disease [3,4,5]. Despite its rarity, intracranial involvement is clinically significant because even small cysts can lead to severe neurological deficits, increased intracranial pressure, and life-threatening complications. Moreover, its nonspecific presentation and radiological resemblance to other intracranial cystic lesions frequently delay diagnosis, making early recognition particularly important in endemic regions [4,5,6,7]. Recent molecular studies have revealed that the clinical presentation and organ tropism of *E. granulosus s.l.* can differ according to genotype. In particular, several authors have suggested an association between atypical localizations and specific genotypes. Some reports have linked intracranial CE to genotype G6 (camel strain), suggesting that certain genotypes may be more susceptible to haematogenous dissemination beyond the liver and lungs [8,9]. The studies concluded that the majority of intracranial CE cases were located in the parietal lobe [5,6,10]. On the other hand, 50% to 75% of intracranial CE is seen in children [4]. The clinical features are largely non-specific and vary depending on the location and severity of the disease. The disease’s clinical presentation often includes symptoms and signs of intracranial hypertension [10,11]. The primary treatment is surgery, but complications such as cyst rupture and/or recurrence during or after surgery can affect patients’ prognoses. Furthermore, surgical treatment is not applicable to all patient groups. Therefore, anthelmintic drugs such as benzimidazoles are used to complement treatment in cases of cyst rupture and/or recurrence [6,10,11].

This study aims to analyze demographic, clinical, diagnostic, and treatment data on intracranial CE cases worldwide over the past 35 years (1990–2024), providing a more comprehensive understanding of the disease.

## 2. Materials and Methods

### 2.1. Search Strategy

This systematic study was conducted according to the Preferred Reporting Items for Systematic Reviews and Meta-Analysis (PRISMA) guidelines [12]. The study review protocol was registered in the Prospective Register of Systematic Reviews (PROSPERO) with [CRD 42024608624].

This search in literature was carried out in PubMed, Scopus, and Web of Science (WoS) databases published between 1990 and 2024 (last updated date: 1 April 2025). Various keyword combinations with Boolean operators were used to find articles involving CE of the brain, including terms such as “brain”, “cerebral”, “central nervous system*”, “intracranial”, and “*Echinococcus granulosus*”. No language restrictions were applied. DeepL and ChatGPT were used to translate non-English articles. The Appendix A provides comprehensive information covering all aspects of the review process.

### 2.2. Study Selection and Data Extraction

Studies based on individual data were included. Reports were considered eligible if intracranial CE cases were confirmed either in the laboratory (via microscopic examination, or molecular testing) or through imaging techniques. The exclusion criteria were as follows: articles not related to intracranial CE; experimental or veterinary studies; studies with insufficient data; commentaries; books/chapters; conference proceedings; reviews without original data; duplicate data; articles lacking full-text access; articles related to *Echinococcus multilocularis* (Alveolar Echinococcosis); and serological or community-based studies.

This search was conducted using the PubMed, Scopus, and WoS databases and all records exported to EndNote v20.5. After removing duplicates, the data was transferred to Excel (Microsoft Excel-files, v2016). The intracranial CE search words were scanned in the titles and abstracts of the articles. Two reviewers assessed the full-text articles to determine final eligibility, resolving any contradictions among studies through discussion and consensus. One author extracted the required data, which was then cross-checked by a second author. Additionally, the reference lists of selected full-text papers were manually examined to identify articles not retrieved by the database search.

Information extracted from these reports included patients’ demographic details such as age, sex, region of origin, medical history, presenting signs and symptoms, diagnostic procedures, the site and size of the cyst(s), therapeutic interventions, clinical outcomes, and follow-up details.

### 2.3. Risk of Bias (Quality) Assessment

Quality assessment was performed using The Joanna Briggs Institute (JBI) critical appraisal tools [https://jbi.global/critical-appraisal-tools (accessed on 1 May 2025)] on each study by two investigators.

### 2.4. Statistical Analysis

The extracted data was entered into Microsoft Excel-files (v2018) and later transformed into Statistical Package for the Social Sciences (SPSS) files (v20.0) for analysis. The findings were presented as descriptive statistics. Categorical variables were summarized using frequencies and percentages, while continuous variables were reported as mean, standard deviation (SD), median, and interquartile range (IQR).

## 3. Results

### 3.1. Study Characteristics

A total of 4642 records were identified through the database search. Of these, 3171 records were excluded prior to screening due to duplication and irrelevant data. Among the remaining 1471 records, 244 were excluded based on the eligibility criteria, and the full text of 82 records could not be accessed. After screening 1145 reports, 376 studies were included. Additionally, 16 reports were retrieved from the reference lists and included in the review. As a result, a total of 392 reports were included in this systematic review (Figure 1).

A total of 718 cases from 392 reports, published between 1990 and 2024 across 40 different countries, are included in this study. Among the included 392 articles, 290 (73.8%) were case reports. The highest number of publications on this topic was recorded in 2019. The leading countries contributing to the study were Türkiye, with 128 studies (32.6%), and India, with 82 studies (20.9%) (Figure 2).

### 3.2. Clinical Features and Diagnostic Characteristics

The mean age of the 718 cases examined in our study was 17.9 ± 14.3 years. Of the 711 cases for which gender was specified, 421 (59.2%) were male, 290 (40.8%) were female, and seven were unspecified. Of these cases, 467 (65.0%) were under 18 years of age and 251 (35.0%) were aged 18 or over.

Of the 718 cases of intracranial CE, 680 (94.7%) presented with symptoms, while 38 (5.3%) did not present with symptoms or these were not reported. Of those with symptoms, the most common findings were signs of increased intracranial pressure in 540 patients (79.4%), motor and neurological deficits in 258 patients (37.9%), visual abnormalities in 158 patients (23.2%), and seizures/convulsions in 147 patients (21.6%). Of the 540 patients with increased intracranial pressure, 396 experienced headaches, 241 had nausea and vomiting, and 190 had papilledema. In 88 patients, no specific symptoms such as headache, nausea/vomiting, or papilledema were reported; however, they were described as having signs of increased intracranial pressure and were therefore included among the 540 cases with increased intracranial pressure. The criteria used to define intracranial pressure increase in these patients were not clearly specified (Table 1).

Among the symptoms evaluated under the heading motor neurological deficits and findings, hemiparesis/hemiplegia was the most common, occurring in 215 cases (31.6%). Other findings evaluated under this heading included paraplegia, increased/decreased deep tendon reflexes, muscle weakness, and tremor. Under the heading of coordination and balance disturbances, ataxia was the most prevalent symptom, occurring in 70 cases (10.2%). In this group, cerebellar signs such as dysmetria, dysdiadochokinesia, and loss of balance, as well as other balance-related findings such as nystagmus and vertigo, were also evaluated.

Cranial CT and MRI were the most commonly used imaging methods, accounting for 80.5% and 60.1% of cases, respectively. Other imaging methods included EEG, MR spectroscopy, and ventriculography, and among these, angiography was the most common in 12 cases (1.8%). Of the cases, 111 (29.4%) had cysts smaller than 5 cm and 266 (70.6%) had cysts measuring 5 cm or larger. The largest cyst measured 170 mm × 150 mm × 150 mm and was located in the temporoparietooccipital region. The majority of cysts (69.2%) were single cysts (Table 1). Data on the status of 568 cysts were available at the time of diagnosis. While 73 cysts (12.9%) were ruptured, 476 (83.8%) were not. Nineteen cysts (3.3%) were reported to be calcified.

Risk factors were reported for 128 cases, and the most common risk factor was found to be living in rural areas, with 50 people (39.1%) identifying risk factors such as animal husbandry, contact with dogs, and farming.

### 3.3. Anatomical Locations

A total of 798 cysts with known localization were identified. Of these, 635 (79.6%) were single and 163 (20.4%) were multiple. The supratentorial area was found to contain 710 cysts (88.9%), while the infratentorial area was found to contain 88 cysts (11.1%). Of the 718 cases examined in our study, 635 (88.4%) had a single cyst and 83 (11.6%) had multiple cysts. The localizations of 163 multicysts from 65 patients were reported. Multiple cysts from 18 patients whose anatomical locations could not be specified due to their large number or widespread distribution were excluded from the calculation (Table 2).

Supratentorial cysts were most commonly found in the parietal region of the cerebral hemisphere, accounting for 132 cases (16.5%). No detailed localization was provided for 18 cysts (2.2%), but only the supratentorial region was reported. Similarly, for 36 (4.6%) of the cysts detected in the infratentorial region, no detailed localization was provided and only the posterior fossa was reported. Of the cysts with a detailed localization, 25 cases (3.1%) were concentrated in the cerebellum (Table 2).

Data were available for 412 patients with extracranial hydatid cysts. Of these patients, extracranial cysts were detected in 137 (33.3%), while no cysts were detected in the remaining 275 patients (66.7%). A total of 222 extracranial hydatid cysts were detected in 137 patients. The most commonly affected organs were the liver (80 cysts, 36.0%), followed by the lungs and heart (37 cysts each, 16.7%). These were followed by the kidney (23 cysts, 10.4%) and muscle tissue (12 cysts, 5.4%). The “others” group comprised 33 cases (14.9%), involving other locations such as the spleen, bone, artery, mediastinum, and spinal canal.

### 3.4. Overview of Treatment Methods for Intracranial Hydatid Cysts

In this review, treatment information was available for 665 patients (92.6%), while 53 patients did not have this information. Of those treated, 637 (95.8%) underwent surgical/interventional treatment, while 28 (4.2%) received medical treatment alone. Of the patients who received only medical treatment, three refused surgical treatments and eight were considered high-risk or inoperable due to comorbidities or cyst location. The most commonly used surgical technique was Dowling’s technique (Table 3).

Albendazole was the most commonly used treatment for both preoperative and postoperative medical care. The median duration of preoperative treatment was 3 months (interquartile range (IQR): 3–3 months), and the median duration of postoperative treatment was 3 months (IQR: 2–6 months). Of the 86 patients who received both preoperative and postoperative treatment, 81 preferred albendazole for both. The median duration of medical treatment alone was five months (IQR: three to six months). Albendazole was also the most commonly used single-agent treatment for medical treatment alone (Table 3).

### 3.5. Clinical Outcome and Follow-Up

An analysis of the clinical outcomes of patients diagnosed with intracranial cysts revealed that 461 (85.5%) of the patients for whom data were available recovered completely, 31 (5.8%) recovered with sequelae, and 47 (8.7%) cases resulted in death. Of the 31 patients who recovered with sequelae, 10 had epilepsy, 9 had hemiparesis/hemiplegia, 8 had a visual impairment, 2 had cranial nerve damage, and 2 experienced headaches. Of the 47 patients who died, complications such as cyst-related rupture, septicemia, and concurrent secondary cysts were cited in 12 patients; recurrence in 5 patients; surgical complications in 4 patients; and missed diagnosis in 2 patients. The cause of death was not stated for 24 patients (Table 4).

A follow-up period of 295 cases was reported, and the median follow-up period for these cases was calculated as 12 months (interquartile range: 6–24 months). While no postoperative recurrence or relapse was observed in 74.0% of the 265 cases, recurrence or relapse was reported in 26% of cases (Table 4).

## 4. Discussion

The presence of multiple intracranial hydatid cysts is usually caused by the spontaneous, traumatic, or accidental rupture of a cyst (during surgical removal), or several protoscolices can spread through the cranial area haematogenously from the liver or lungs [1,2,13]. A large proportion of the oncospheres (approximately 75%) localize in the liver, while some oncospheres (20%) that pass through the hepatic filter settle in the lungs. More rarely, via systemic circulation, they may spread to other organs—particularly the spleen, kidneys, bones, and brain [2,13]. Approximately 2% of oncospheres, after entering systemic circulation, can reach the brain through haematogenous spread and cause intracranial CE [3,7]. Although genotype data were not available for the majority of the included cases, the potential role of *E. granulosus s.l.* genotypes in determining organ involvement, including intracranial localization, has been emphasized in previous molecular studies. Several reports have described an association between atypical localizations, including brain involvement, and the G6 genotype. It has been suggested that this genotype exhibits distinct biological behavior, which could lead to systemic dissemination and increase the likelihood of extrahepatic organ involvement [8,9]. However, the absence of genotype reporting in most published cases limits our ability to analyze this relationship in the present review.

The endemic regions are Central Asia, Mediterranean countries, South America, and East Africa [1,2,14]. In this review, CE was found to be more frequently reported from Türkiye and India. CE is commonly seen in children and young adults. This might be due to the more occupational status of males than females working on agriculture, hunting in the rural area where they are in close proximity to dogs, wolves, or foxes in endemic areas [1,2,3].

In this review, 392 publications reporting a total of 718 intracranial CE cases were included. Most patients were children (65%) and male (59.2%). The most common clinical presentation was features of raised intracranial pressure (79.4%), followed by motor deficits (37.9%) and visual symptoms (23.2%). Imaging data showed that the majority of cysts were supratentorial (88.9%), particularly in the parietal lobe, and that most patients had a single cyst (88.4%). Surgical management was undertaken in nearly all cases (95.8%), often accompanied by albendazole therapy. Outcomes were generally favourable, with 85.5% of patients achieving full recovery. Nevertheless, 8.7% of patients died, primarily due to complications related to rupture. Information from follow-ups was available for 26% of cases.

The symptoms and signs of CE depend on the location of the cyst. In intracranial CE, children are mostly presented with signs of raised intracranial pressure such as headache, nausea, vomiting, and papillae edema. Focal neurological signs such as motor weakness, speech disorders, disturbances of sensation, visual field disorders, and epilepsy are usually seen in adults [3,7]. Consistent with previously published data, increased intracranial pressure was also the most common presenting feature in this study, reported in 79.4% of cases with symptoms such as headache, nausea/vomiting, or papilledema. The second most common group of symptoms were focal neurological findings (37.9%), while epileptic seizures and visual disturbances were reported in 21.6% and 23.2% of cases, respectively. These rates suggest that intracranial CE may present with a broad range of focal neurological manifestations, not just signs of raised intracranial pressure, and highlight the need for clinical awareness in its diagnosis and management.

In the literature, the most common neurological investigations are CT and MRI. With the advances in MRI, diagnostic accuracy has improved. It plays a fundamental role in early diagnosis. MRI findings of CE are often a spherical, round cyst with cyst fluid darker than CSF intensity on T1-weighted images with no contrast enhancement. Differential diagnosis is mainly other space-occupying cystic lesions such as arachnoid cyst, cystic glial tumors, and brain abscess [5,15,16]. In line with the existing literature, these findings confirm that CT and MRI were the primary imaging modalities used in the reported cases. CT was performed on 80.5% of patients and MRI on 60.1%. The detection of antibodies and antigens are used to support the diagnosis, and the detection of antigens is less sensitive [13,16]. As shown in this review, serology was positive in 56.6% of cases.

Intracranial CE is found in 3 forms: intracerebral, extracerebral, and combined. The intracerebral form is mostly located supratentorially, in the territory of the middle cerebral artery, mainly in the parietal lobe. Giant cysts are multilobar. The cysts are often solitary; rarely, there are multifocal cysts [6,10,11,17]. In this data, 88.4% of them were single, and 88.9% of all of them were located in supratentorial region. These localization features also influence the choice of treatment approach.

Commonly, the cases are presented with the signs and symptoms of raised intracranial pressure or focal neurological deficits that must be treated as soon as possible to avoid deterioration or cerebral herniation. There can be no time for waiting for the possible effects of medical treatment. Surgery becomes the emergency intervention. Radical surgical removal of the cyst using Dowling’s hydro-dissection technique is the gold standard, whereas the puncture–aspiration–injection–reaspiration (PAIR) technique followed by medical therapy serves as the main alternative when surgery is not feasible.If there are contraindications of surgery or the prediction of a poor surgical outcome, medical treatment is the first choice [6,18]. To avoid surgical complications, it is essential to evaluate radiological examinations preoperatively for planning the site and size of craniotomy properly [16]. This review revealed that the treatment approaches documented in the literature largely aligned with clinical practice. Surgical or interventional management was performed on 95.8% of patients for whom data was available, confirming that surgery remains the primary treatment for intracranial CE. Consistent with previous studies, Dowling’s technique was the most frequently used surgical method. Only a small proportion of patients (4.2%) received medical therapy alone, primarily due to surgical refusal, high operative risk, or unsuitable cyst location. These findings align with previous reports emphasizing the limited indications for non-surgical management. Albendazole was the most frequently used antiparasitic drug in preoperative, postoperative, and standalone medical treatments. Although factors such as cyst size or rupture could theoretically influence the outcome of medical treatment, the included studies—predominantly single-case reports with limited or absent follow-up—did not provide sufficient data to reliably assess the treatment response or clarify the reasons for potential albendazole failure. These findings reinforce the idea that, while radical surgical excision remains the preferred approach, individualized treatment decisions are required in cases involving challenging anatomical locations, comorbidities, or contraindications to surgery.

In general, the successful surgical removal of single cyst followed by medical treatment has a very good outcome. Preoperatively spontaneous or intraoperatively accidental rupture of the cyst can lead to anaphylactic shock, focal neurological deficits, hydrocephalus, intracranial haemorrhage, and stroke that cause poor prognosis [5,11,19]. In our analyses of 718 cases, complete recovery was achieved in 85.5% of them, while the rate of neurological sequel was found to be 5.8%. In the literature, postoperative mortality rates of the disease have been reported to range between 3% and 21% [20,21,22,23]. Among the causes of death, anaphylactic shock due to intraoperative cyst rupture and postoperative infections are the most prominent [10,22,24]. In this study, we found the mortality rate to be 8.7%. The main causes were anaphylactic shock and septicemia.

In this data of 718 cases, the follow-up period was mentioned only in 295 cases (41%). Among them, the recurrence rate was 26%. In the literature, recurrence rates have been reported to range from 8% to 37.5%. Most recurrences are associated with intraoperative cyst rupture, which underscores the importance of surgical techniques [6,19,25]. In this review as well, the recurrence rate due to intraoperative cyst rupture was the same. Follow-up periods vary among studies, with some cases showing long-term follow-up ranging from 8 to 45 years [23,26,27]. These extended follow-up periods provide valuable data for evaluating the disease course and treatment efficacy. In this study, the mean follow-up period was 12 months.

## 5. Limitations of Our Study

This study has certain limitations. First, most of the included studies consisted of case reports or small case series, leading to heterogeneity of the data and limiting the possibility of robust statistical analyses. Second, follow-up periods varied considerably between studies, and long-term outcomes were not available for many cases. Lastly, a large proportion of the included cases originated from regions such as Türkiye, India, and Iran. These regions have higher rates of cystic echinococcosis. This geographical clustering may have influenced the observed patterns. These included symptoms, diagnostics, and management. This potentially limits the generalizability of the findings.

## 6. Conclusions

In this current study, we aimed to analyze demographic, clinical, diagnostic, and treatment data on intracranial CE cases worldwide over the past 35 years (1990–2024), providing a more understandable definition of CE. It was demonstrated that intracranial CE usually affects developing countries, and the predominance is particularly notable among younger males.. Surgery should be considered as the first and definitive treatment of choice, followed by antihelmintic medical treatment. Although cerebral involvement is rare, intracranial CE, which is more frequently encountered, especially in the paediatric age group, should always be considered in the differential diagnosis in endemic regions.

## Figures and Tables

**Figure 1 pathogens-14-01264-f001:**
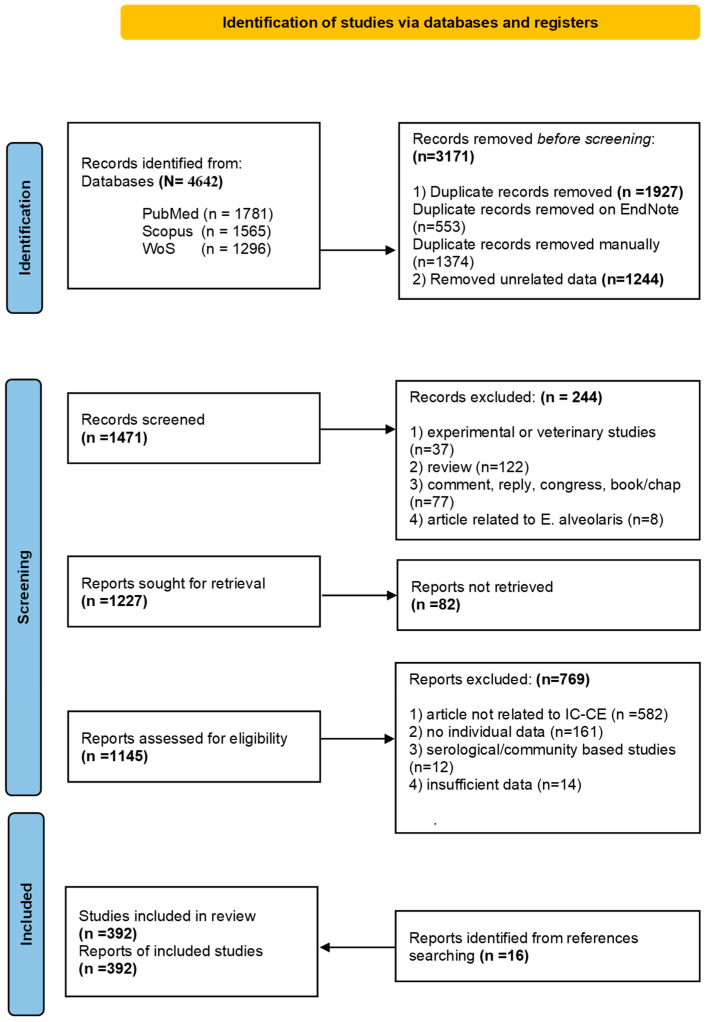
PRISMA 2020 flow diagram for new systematic reviews that included searches of databases and registers only [12].

**Figure 2 pathogens-14-01264-f002:**
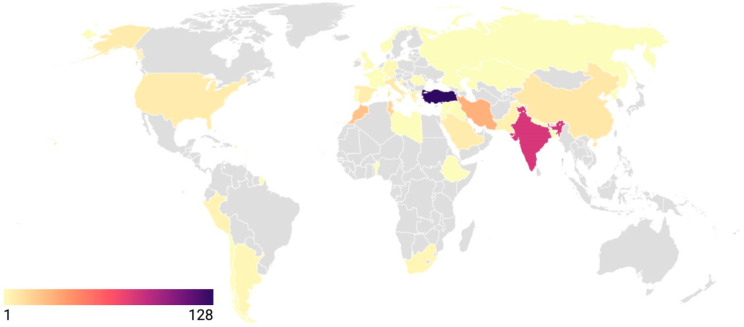
Geographical distribution of intracranial cystic echinococcosis studies by country. The map is adapted from Datawrapper [https://www.datawrapper.de (accessed on 15 October 2025)].

**Table 1 pathogens-14-01264-t001:** Clinical features and diagnostic characteristics.

	Number of Cases (%)		
	All Cases (N = 718)	Adult Patients (n = 251)	Paediatric Patients * (n = 467)
Age Mean ± SD	17.9 ± 14.3	33.5 ± 13.5	9.5 ± 3.7
Gender (No Data)	7	3	4
Female Male	290 (40.8) 421 (59.2)	104 (41.9) 144 (58.1)	186 (40.2) 277 (59.8)
Symptoms and Signs **	(n = 680)	(n = 242)	(n = 438)
Increased Intracranial Pressure Signs Headache Nausea and Vomiting Papilledema Motor Neurological Deficits and Findings Visual Abnormalities Seizure/Convulsion Coordination and Balance Disturbances Altered Consciousness Cognitive or Mental Status Alterations Speech and Language Disorders Others	540 (79.4) 396 (58.2) 241 (35.6) 190 (27.9) 258 (37.9) 158 (23.2) 147 (21.6) 99 (14.6) 66 (9.7) 41 (6.0) 37 (5.4) 79 (11.6)	190 (78.5) 171 (70.7) 88 (36.4) 45 (18.6) 85 (35.1) 54 (22.3) 58 (24.0) 43 (17.8) 28 (11.6) 20 (8.3) 12 (5.0) 31 (12.8)	350 (79.9) 225 (51.4) 153 (34.9) 145 (33.1) 173 (39.5) 104 (23.7) 89 (20.3) 56 (12.8) 38 (8.7) 21 (4.8) 25 (5.7) 48 (11.0)
Serology	(n = 168)	(n = 69)	(n = 99)
Positive Negative	95 (56.6) 73 (43.4)	49 (71.0) 20 (29.0)	46 (46.5) 53 (53.5)
Eosinophilia	(n = 82)	(n = 34)	(n = 48)
Positive Negative	33 (40.2) 49 (59.8)	16 (47.1) 18 (52.9)	17 (35.4) 31 (64.6)
Imaging Modalities Performed **	(n = 671)	(n = 247)	(n = 424)
Cranial CT Cranial MRI Cranial Plain Radiography Others	540 (80.5) 403 (60.1) 72 (10.7) 28 (4.6)	196 (79.4) 174 (70.4) 15 (6.1) 6 (2.4)	
Maximum Cyst Diameter at Diagnosis ***	(n = 377)	(n = 117)	(n = 260)
<5 cm ≥5 cm	111 (29.4) 266 (70.6)	47 (40.2) 70 (59.8)	64 (24.6) 196 (75.4)
Cyst Morphology	(n = 705)	(n = 243)	(n = 462)
Single Multiple	497 (70.5) 208 (29.5)	122 (50.2) 121 (49.8)	375 (81.2) 87 (18.8)

* <18 years ** Due to the presence of multiple simultaneous symptoms in individual patients and the concurrent use of more than one imaging modality, the total number of reported symptoms and imaging methods exceeds the number of patients. *** For patients with multiple cysts, the size of the largest cyst was considered.

**Table 2 pathogens-14-01264-t002:** Anatomical locations of cystic echinococcosis.

Region (N = 798)	Anatomical Distribution	Total (%)	Localization (%)		
			Left	Right	No Data
Supratentorial (n = 710)	Cerebral Hemispheres	132 (16.5) 85 (10.7) 59 (7.4) 43 (5.4) 102 (12.8) 99 (12.4) 62 (7.8) 17 (2.1) 4 (0.5) 33 (4.1) 20 (2.5) 5 (0.6) 1 (0.1)	67 41 25 12 58 44 26 9 1 12 14 3 -	52 32 26 22 32 44 27 6 2 18 4 2 1	13 12 8 9 12 11 9 2 1 3 2 - -
	Parietal Frontal Occipital Temporal Frontoparietal * Parietooccipital * Parietotemporal * Frontotemporal * Temporooccipital * Frontotemporoparietal ** Temporoparietooccipital ** Frontoparietooccipital ** Frontotemporoparietooccipital **				
	Ventricular System	24 (3.0) 6 (0.8)	10 -	9 -	5 -
	Lateral ventricle Third ventricle				
	Mentioned as supratentorial	18 (2.2)			
Infratentorial (n = 88)	Mesencephalon Aqueduct of sylvius Pons Fourth ventricle Cerebellum Cerebellopontine angle	2 (0.3) 4 (0.5) 6 (0.8) 6 (0.8) 25 (3.1) 9 (1.1)	- - - - 10 3	- - - - 5 3	- - - - 10 3
	Mentioned as infratentorial	36 (4.6)			

* Involving two lobes; ** involving three or more lobes.

**Table 3 pathogens-14-01264-t003:** Overview of treatment methods for intracranial cystic echinococcosis.

	Number of Cases (%)	Duration of Medical Treatment in Months Median (IQR)/n
Treatment Methods (n = 665)		
Surgery only Surgery and medical treatment Medical treatment only	309 (46.5) 328 (49.3) 28 (4.2)	
Preoperative Medical Treatment (n = 106)		3 (3-3)/71
Albendazole monotherapy Mebendazole monotherapy Albendazole + Praziquantel	99 (93.4) 4 (3.8) 3 (2.8)	3 (3-3)/69 3 (3-3)/1 12 (12-12)/1
Postoperative Medical Treatment (n = 297)		3 (2-6)/201
Albendazole monotherapy Mebendazole monotherapy Albendazole + Praziquantel	273 (91.9) 14 (4.7) 10 (3.4)	3 (2-6)/186 12 (1-12)/7 2 (1-5.25)/8
Medical Treatment Only (n = 28)		5 (3-6)/11
Albendazole monotherapy Mebendazole monotherapy Albendazole + Praziquantel	23 (82.1) - 5 (17.9)	5 (3-6.75)/10 - 6 (6-6)/1

**Table 4 pathogens-14-01264-t004:** Clinical outcome, follow-up, and recurrence/relapse rates.

	Number of Cases	Clinical Outcome				Follow-Up Duration in Months		Postoperative Recurrence		
		Complete Recovery	Sequelae	Death	No Data	Median IQR	No Data	+	−	No Data
All Cases	718	461 (85.5)	31 (5.8)	47 (8.7)	179	12 (6–24)	423	69 (26.0)	196 (74.0)	453
Adults	251	152 (88.4)	5 (2.9)	15 (8.7)	79	12 (6–24)	147	26 (38.8)	41 (61.2)	184
Paediatric	467	309 (84.2)	26 (7.1)	32 (8.7)	100	12 (7–36)	276	43 (21.7)	155 (78.3)	269

## Data Availability

All data generated or analyzed during this study are included in this published article and its Appendix A.

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
