# Peer review of "Epidemiological, Diagnostic, and Clinical Features of Intracranial Cystic Echinococcosis: A Systematic Review"

_pathogens, 2025, doi:10.3390/pathogens14121264_

Round 1

Reviewer 1 Report

Comments and Suggestions for Authors

In the manuscript submitted by Can et al., the authors efficiently review atypical cases of intracranial cystic echinococcosis (CE). However, several aspects should be addressed to reach the quality standards required for publication in this prestigious journal.

  1. It is unfortunate that the item “species/genotypes” was not included in the meta-analysis. Nevertheless, this aspect should be incorporated and discussed in both the Introduction and Discussion sections, as several authors have suggested that atypical localization could be related to specific Echinococcus genotypes.
  2. Likewise, the possible lack of response to albendazole treatment should be explored in greater depth.
  3. I recommend adding a map figure to illustrate the comprehensive geographic coverage of the analyzed studies and countries included.
  4. I also suggest including figures that clearly demonstrate the relationship between intracranial CE, other localizations, and treatment responses (e.g., ABZ, ABZ+PZQ, etc.).
  5. Finally, it would be valuable to expand the Discussion section to address the potential benefits or drawbacks of pre-surgical treatment.

Line 220: Hydatid Cyst

Line 297: CE

Author Response

Dear Sir/Madam,

We would like to sincerely thank you for the time and effort you have devoted to evaluating our manuscript. Your insightful comments, constructive criticisms, and valuable suggestions have been highly appreciated and have significantly contributed to improving the clarity, quality, and overall rigor of our work. We carefully considered each point raised and have revised the manuscript accordingly. Below, we provide a detailed, point-by-point response to all comments.

Reviewer 1

  1. It is unfortunate that the item “species/genotypes” was not included in the meta-analysis. Nevertheless, this aspect should be incorporated and discussed in both the Introduction and Discussion sections, as several authors have suggested that atypical localization could be related to specific Echinococcus genotypes.

In our review, all included cases were reported as Echinococcus granulosus, but genotype information was not available for the vast majority of studies, most of which were single case reports or small case series. Because molecular characterization was rarely performed or not reported, genotype data could not be incorporated into the analysis.

  1. Likewise, the possible lack of response to albendazole treatment should be explored in greater depth.

We discussed potential factors that may contribute to a lack of response to albendazole treatment, such as cyst size and rupture/secondary cyst formation. However, because many of the included studies were case reports with limited or absent follow-up information, treatment response could not be reliably assessed. Therefore, a deeper analysis of albendazole treatment failure was not feasible based on the available data.

  1. I recommend adding a map figure to illustrate the comprehensive geographic coverage of the analyzed studies and countries included.

In accordance with your recommendation, we have added a map figure illustrating the geographic distribution of all included studies and the countries represented in our analysis.

  1. I also suggest including figures that clearly demonstrate the relationship between intracranial CE, other localizations, and treatment responses (e.g., ABZ, ABZ+PZQ, etc.).

Our study focuses exclusively on intracranial CE, and therefore we do not have data on extracranial localizations to compare with. In addition, information on follow-up duration, recurrence, and medical treatment regimens was reported only in a subset of the included cases (Table 3). Most patients with available follow-up data received both medical and surgical treatment, making it impossible to determine the independent effect or success of individual regimens (e.g., ABZ, ABZ+PZQ). For these reasons, we were unable to generate the comparative figures you suggested.

  1. Finally, it would be valuable to expand the Discussion section to address the potential benefits or drawbacks of pre-surgical treatment.

New sections have been added to the discussion and highlighted accordingly.

Reviewer 2 Report

Comments and Suggestions for Authors

This systematic review on intracranial cystic echinococcosis represents a significant and valuable contribution to the literature, constituting one of the largest compiled case series on this rare condition. The study is well-conceived, addresses a relevant clinical topic, and follows appropriate systematic review methodology.

I am confident that the authors can address these concerns. I recommend a thorough, line-by-line revision followed by re-submission.

For the introduction

The transition from general CE to intracranial CE is somewhat abrupt. Consider adding a bridging sentence to explain why intracranial CE, despite being rare, is clinically significant—e.g., due to its severe neurological consequences and diagnostic challenges.

For the Materials and Methods

The section outlines a systematic review process generally aligned with PRISMA guidelines and PROSPERO registration.

he criteria “articles not related to CE” and “articles not related to intracranial CE” are redundant if the search strategy was specific enough. This could be simplified.

For the Results

This section presents a comprehensive and valuable dataset from a large number of cases. The data on demographics, clinical features, cyst characteristics, and outcomes are highly relevant.

For the Discussion

The Discussion section effectively contextualizes the study’s findings within the existing literature and acknowledges the study’s limitations. The conclusions are somewhat vague and do not fully capitalize on the study’s findings.

The discussion jumps between topics (pathogenesis -> epidemiology -> symptoms -> diagnosis -> anatomy -> treatment -> outcomes) without smooth transitions. A more logical structure would be:

  1. Summary of main findings.
  2. Interpretation and comparison with literature.
  3. Clinical implications and recommendations.
  4. Limitations.
  5. Conclusion.

Line76      non-English language articles----non-English articles.

Line100    tolls    tools

Line260   peroperatively-------intraoperatively

Author Response

Dear Sir/Madam,

We would like to sincerely thank you for the time and effort you have devoted to evaluating our manuscript. Your insightful comments, constructive criticisms, and valuable suggestions have been highly appreciated and have significantly contributed to improving the clarity, quality, and overall rigor of our work. We carefully considered each point raised and have revised the manuscript accordingly. Below, we provide a detailed, point-by-point response to all comments.

Reviwer-2

  1. The transition from general CE to intracranial CE is somewhat abrupt. Consider adding a bridging sentence to explain why intracranial CE, despite being rare, is clinically significant—e.g., due to its severe neurological consequences and diagnostic challenges.

We have added a bridging sentence in the Introduction to explain that, although intracranial CE is rare, it is clinically significant due to its potentially severe neurological consequences and diagnostic challenges.

  1. For the Materials and Methods: The section outlines a systematic review process generally aligned with PRISMA guidelines and PROSPERO registration. The criteria “articles not related to CE” and “articles not related to intracranial CE” are redundant if the search strategy was specific enough. This could be simplified.

We have simplified the exclusion criteria as recommended and removed redundant items to improve clarity.

  1. For the Results: This section presents a comprehensive and valuable dataset from a large number of cases. The data on demographics, clinical features, cyst characteristics, and outcomes are highly relevant.
  2. For the Discussion: The Discussion section effectively contextualizes the study’s findings within the existing literature and acknowledges the study’s limitations. The conclusions are somewhat vague and do not fully capitalize on the study’s findings. The discussion jumps between topics (pathogenesis -> epidemiology -> symptoms -> diagnosis -> anatomy -> treatment -> outcomes) without smooth transitions. A more logical structure would be: Summary of main findings. Interpretation and comparison with literature. Clinical implications and recommendations. Limitations. Conclusion.

New sections have been added to the discussion and highlighted accordingly.

  1. Line76      non-English language articles----non-English articles.

It has been corrected.

  1. Line100    tolls    tools

It has been corrected.

  1. Line260   peroperatively-------intraoperatively

It has been corrected.

Round 2

Reviewer 1 Report

Comments and Suggestions for Authors

Es lamentable que el elemento "especie/genotipo" no se incluyera en el metanálisis. Sin embargo, este aspecto debería incorporarse y discutirse tanto en la Introducción como en la Discusión, ya que varios autores han sugerido que la localización atípica podría estar relacionada con genotipos específicos de  Echinococcus granuslosus sl.

Author Response

Dear Sir/Madam,

We would like to sincerely thank you for the time and effort. 

Es lamentable que el elemento "especie/genotipo" no se incluyera en el metanálisis. Sin embargo, este aspecto debería incorporarse y discutirse tanto en la Introducción como en la Discusión, ya que varios autores han sugerido que la localización atípica podría estar relacionada con genotipos específicos de  Echinococcus granuslosus sl.

In accordance with suggestion, we have now incorporated this aspect into both the Introduction and the Discussion.
